# Antioxidative Self-Assembling Nanoparticles Attenuate the Development of Steatohepatitis and Inhibit Hepatocarcinogenesis in Mice

**DOI:** 10.3390/antiox11101939

**Published:** 2022-09-28

**Authors:** Takahisa Watahiki, Kosuke Okada, Ikuru Miura, Keii To, Seiya Tanaka, Eiji Warabi, Naomi Kanno, Kenji Yamagata, Naohiro Gotoh, Hideo Suzuki, Shunichi Ariizumi, Kiichiro Tsuchiya, Yukio Nagasaki, Junichi Shoda

**Affiliations:** 1Department of Gastroenterology, Faculty of Medicine, University of Tsukuba, 1-1-1 Tennodai, Tsukuba 305-8575, Japan; 2Doctoral Program in Clinical Sciences, Graduate School of Comprehensive Human Sciences, University of Tsukuba, 1-1-1 Tennodai, Tsukuba 305-8575, Japan; 3Division of Medical Sciences, Faculty of Medicine, University of Tsukuba, 1-1-1 Tennodai, Tsukuba 305-8575, Japan; 4Doctoral Program in Sports Medicine, Graduate School of Comprehensive Human Sciences, University of Tsukuba, 1-1-1 Tennodai, Tsukuba 305-8575, Japan; 5Doctoral Program in Medical Sciences, Graduate School of Comprehensive Human Sciences, University of Tsukuba, 1-1-1 Tennodai, Tsukuba 305-8575, Japan; 6Department of Food Science and Technology, Tokyo University of Marine Science and Technology, 4-5-7 Konan, Minato-ku, Tokyo 108-8477, Japan; 7Division of Biomedical Sciences, Faculty of Medicine, University of Tsukuba, 1-1-1 Tennodai, Tsukuba 305-8575, Japan; 8Department of Oral and Maxillofacial Surgery, Institute of Clinical Medicine, Faculty of Medicine, University of Tsukuba 1-1-1 Tennodai, Tsukuba 305-8575, Japan; 9Department of Surgery, Institute of Gastroenterology, Tokyo Women’s Medical University, 8-1 Kawada-cho, Shinjuku-ku, Tokyo 162-8666, Japan; 10Faculty of Pure and Applied Sciences, University of Tsukuba, 1-1-1 Tennodai, Tsukuba 305-8573, Japan

**Keywords:** hepatocellular carcinoma, HCC, nonalcoholic steatohepatitis, NASH, nuclear-factor-erythroid-2-related factor 2, NRF2, p62/sequestosome 1, p62, antioxidative self-assembling nanoparticles, SMAPo^TN^

## Abstract

Oxidative stress (OS) contributes to nonalcoholic steatohepatitis (NASH) and hepatocarcinogenesis. We investigated whether antioxidative self-assembling nanoparticles (SMAPo^TN^) could reduce the development of NASH and hepatocellular carcinoma (HCC) in *p62/Sqstm1* and *Nrf2* double knockout (DKO) mice and studied protective mechanisms. We measured disease development in male DKO mice fed a normal chow (NASH model) or a 60% high-fat diet (HFD; HCC model) with or without SMAPo^TN^ administration for 26 weeks. SMAPo^TN^ inhibited liver fibrosis in both groups and prevented HCC development (0% vs. 33%, *p* < 0.05) in the HFD group. SMAPo^TN^ reduced OS, inflammatory cytokine signaling, and liver fibrosis. RNA-sequencing revealed that SMAPo^TN^ decreased endoplasmic reticulum stress signaling genes in both groups, HCC driver genes, and cancer pathway genes, especially PI3K-AKT in the HFD groups. In the SMAPo^TN^ treatment HFD group, serum lipopolysaccharide levels and liver lipopolysaccharide-binding protein expression were significantly lower compared with those in the nontreatment group. SMAPo^TN^ improved the α-diversity of gut microbiota, and changed the microbiota composition. Oral SMAPo^TN^ administration attenuated NASH development and suppressed hepatocarcinogenesis in DKO mice by improving endoplasmic reticulum stress in the liver and intestinal microbiota. SMAPo^TN^ may be a new therapeutic option for NASH subjects and those with a high HCC risk.

## 1. Introduction

Obesity causes numerous metabolic disorders and complications, including cardiovascular disease, diabetes mellitus type 2, and nonalcoholic fatty liver disease (NAFLD) [1,2]. Nonalcoholic steatohepatitis (NASH) is a progressive liver disease characterized by steatosis, inflammation, and fibrosis, leading to liver cirrhosis and hepatocellular carcinoma (HCC) [2,3]. The development of NASH and HCC is associated with multiple parallel factors, and this theory is called the “multiple parallel hits theory” [4]. Oxidative stress (OS) [5,6], endoplasmic reticulum (ER) stress [7], lipopolysaccharide (LPS) derived from the intestines [8,9], and insulin resistance [3] have been reported to be associated with hepatocarcinogenesis through changes in several cancer driver genes and cancer pathway genes [10,11]. However, the molecular mechanisms remain unclear. In addition, several pharmacological strategies that have been tested for the treatment of NASH-based fibrosis are not yet ready for clinical use [12,13], and no preventive therapy has been developed.

In NASH, the accumulation of certain lipid species injures hepatocytes via lipotoxicity [14], which generates reactive oxygen species (ROS) and induces hepatocyte cell death, liver inflammation, and liver fibrosis, thereby contributing to hepatocarcinogenesis [15,16]. Given the importance of OS in NASH development, antioxidant interventions have been tested in humans with mixed results [17,18]. Moreover, because NASH is often triggered by the disruption of the gut–liver axis, including intestinal dysbiosis and intestinal barrier dysfunction [19], the pharmacological treatment of NASH requires strategies based on the gut–liver axis.

In recent years, patients with NASH have increased in several countries, such as European countries, the United States, China, and Japan, and the number of NASH subjects with advanced fibrosis (F3 and F4 stages) who are likely to develop HCC is predicted to increase to approximately one million in Japan by 2030 [20].

In addition, the proportion of HCC patients with nonviral etiologies has remarkably increased from 10.0% in 1991 to 32.5% in 2015 in Japan [21]. NAFLD, including NASH, accounts for 15.1% of non-B and non-C HCC background liver disease. Moreover, in recent years, the proportion of patients with diabetes mellitus, obesity, dyslipidemia, hypertension, and fatty liver has been increasing in Japan. Therefore, NASH-related HCC is predicted to further increase in the future [21]. Accordingly, developing new therapeutic options for NASH and NASH-related HCC is an urgent issue.

Oxidative stress and lipid peroxidation, which are caused by an increased production of reactive oxygen species (ROS), induce NASH with gut-derived endotoxins and/or any number of cytokines such as tumor necrosis factor-α (TNFα) and adipocytokines [22]. The action of these factors results in liver injury and subsequent progression towards hepatic fibrosis [23]. Antioxidants are one of the potential candidates for the amelioration of OS in NAFLD, NASH, and HCC. However, most of the conventional antioxidants with a low molecular weight, such as 4-hydroxy-2, 2, 6, and 6-tetramethylpiperidine-l-oxyl (TEMPOL), do not exert suitable effects [24]. TEMPOL degrades superoxides and peroxides directly in vitro [25]. These conventional antioxidants spread throughout the whole body and are eliminated rapidly [26]. Additionally, these antioxidants disrupt the normal redox reaction in normal cells, including the electron transport chain, leading to mitochondrial dysfunction and apoptosis [27,28]. Therefore, these antioxidants have a very small or no “therapeutic window”. One of the authors and his colleagues have recently started developing nanoparticle-type self-assembling antioxidants, which stay for an extended period of time in the intestinal mucosa after oral administration. Since the size of several tens of nanometers suppresses internalization in normal cells to avoid the dysfunction of intracellular redox reactions and decreases the unwanted adverse effects, as a result, these nanoparticle-type antioxidants expand their therapeutic window significantly. For example, one of these nanoparticle-type antioxidants, RNP^O^, which does not disintegrate regardless of environmental pH changes, has been shown to inhibit the increase in commensal bacteria in the colonic mucosa of a colitis mouse model by scavenging ROS and suppressing inflammation [29]. Another type of antioxidant nanoparticle, RNP^N^, which disintegrates under acidic environments, is absorbed from the intestinal tract, remains in the liver, and exerts anti-inflammatory and antifibrosis activity with anti-OS effects in mild NASH induced by choline-deficient l-amino-acid-defined feeding [30]. However, whether SMAPo^TN^ prevents NASH-related HCC has not been tested.

In previous studies, genetically modified mice and/or carcinogen-induced models, such as phosphatase and tensin homolog-deficient mice [31] and mice with HCC induced through diethylnitrosamine [32], were examined as HCC models. We developed *p62/sequestosome 1 (p62)* and *nuclear-factor-erythroid-2-related factor 2 (Nrf2)* double knockout mice (DKO); male DKO mice spontaneously developed adult-onset NASH and its associated hepatocarcinogenesis following normal chow (NC) feeding [33], exhibiting a phenotype similar to human NASH [33]. Because of the phenotypic similarities to the clinical features of human NASH, such as obesity (visceral fat accumulation) coupled with metabolic syndrome, insulin resistance, and adipokine imbalance, this DKO mouse represents a unique animal model for exploring novel therapeutic approaches for the prevention and/or treatment of NASH. More importantly, the DKO mice show dysbiosis associated with an increased proportion of Gram-negative bacteria species and elevated LPS levels in feces. Collectively, in the DKO mice, the activation of innate immune responses through excessive LPS flux from the intestines, occurring both within and outside the liver, is central to the development of hepatic damage in the form of NASH. The DKO mice are a suitable model for investigating the therapeutic effect of antioxidative nanoparticle on NASH and NASH-related HCC.

The aim of the present study was to elucidate whether SMAPo^TN^ exerts preventive effects against the development of NASH and its associated hepatocarcinogenesis and clarify the protective mechanisms by which the oral administration of SMAPo^TN^ attenuates NASH development and suppresses hepatocarcinogenesis through the gut–liver axis in DKO mice.

## 2. Materials and Methods

### 2.1. Animals

*p62* and *Nrf2* DKO mice were produced by crossing *p62*-KO and *Nrf2*-KO mice and regenotyping [33]. All mice were maintained under specific pathogen-free conditions in an environmentally controlled clean room at the Laboratory Animal Resource Center, University of Tsukuba. Mice were housed at an ambient temperature of 23 ± 2.5 °C with a daily light/dark cycle from 05:00 to 19:00. All experiments were performed under protocols approved by the Institutional Animal Care and Use Committees of the University of Tsukuba. Male mice were used in all experiments.

The mice were fed a normal chow (NC: 5.1% fat, 23.1% protein, 360 kcal/100 g from Oriental Yeast, Tokyo, Japan) or a 60% high-fat diet (HFD: 60% fat, 24.5% protein, 640 kcal/100 g from Oriental Yeast, Tokyo, Japan) for 26 weeks from 6 to 32 weeks of age. During the same period with HFD feeding, the mice had free access to drinking water with SMAPo^TN^ (10 mg/day) or without SMAPo^TN^.

### 2.2. Preparation and Pharmacokinetics of SMAPo^TN^

A poly (styrene-co-maleic acid) copolymer (PSMA, molecular weight (MW) = 7500, styrene:maleic anhydride unit ratio = 2:1) was dissolved in anhydrous tetrahydrofuran (THF). An alpha-methoxy-omega-hydroxy poly (ethylene glycol) (5000 MW, MeO-PEG-OH) was dissolved in dry THF, and a stoichiometric amount of butyllithium (1.6 M hexane solution) was added to convert the PEG end to an alcoholate. This solution was added to a separately prepared THF solution of PSMA and stirred for 1 h. The reaction solution was poured into isopropyl alcohol to obtain a precipitate. The precipitate was dispersed in hexane, purified through filtration, and dried in vacuo (SMAPo). The synthesized SMAP_O_ was dissolved in anhydrous THF. The excess molar amount of NH_2_-TEMPO versus maleic anhydride repeating units in SMAPo was dissolved in anhydrous THF, added to the above solution, poured into ether, and the precipitate was collected to obtain SMAP_O_^TN^. In this work, one PEG chain was introduced to the PSMA backbone via ester-linkage, and NH_2_-TEMPO was introduced to the remaining maleic anhydride ring (18-TEMPO molecules per PSMA backbone). The pharmacokinetics of SMAPo^TN^ in mice after oral administration were determined using fluorescent rhodamine-labeled SMAPo^TN^. The rhodamine-labeled SMAPo^TN^ was administered orally using a sonde. After oral administration from 1.5 h to 48 h, the labeled SMAPo^TN^ was observed with an Intelligent Visual Information System (IVIS; Perkin Elmer, Waltham, MA, USA). Mice were anesthetized through the inhalation of isoflurane (Perkin Elmer, MA, USA) and subjected to abdominal incisions. All IVIS images were acquired at excitation/emission wavelength 535/580 nm with an exposure time of 1 s. Acquired IVIS images were adjusted to the same minimum and maximum values of the color scale with Living Image Software (Perkin Elmer, MA, USA). The color scale indicated the radiant efficiency of the rhodamine-labeled SMAPo^TN^ with the IVIS (p/s/cm^2^/sr/µW/cm^2^).

### 2.3. Biochemical Analysis

Serum concentrations of aspartate aminotransferase (AST), alanine aminotransferase (ALT), triglycerides (TGs), high-density lipoprotein cholesterol (HDL-CHO), low-density lipoprotein cholesterol (LDL-CHO), and free fatty acids (FFAs) were measured with Oriental Yeast (Tokyo, Japan) using a Hitachi 7180 Auto Analyzer. AST and ALT were measured with the JSCC transferable method using L type Wako AST or ALT-J2 kit (FUJIFILM, Tokyo, Japan). TGs and FFAs were measured with enzymatic methods using L-type Wako TG-M or NEFA HF (FUJIFILM, Tokyo, Japan). HDL-CHO and LDL-CHO were measured with direct methods using CHOLESTEST N LDL or HDL (SEKISUI MEDICAL, Tokyo, Japan). Glutathione (GSH and GSSG) concentrations and superoxide dismutase (SOD) activity in the liver tissue specimens were measured using a GSSG/GSH Quantification Kit (FUJIFILM, Tokyo, Japan) and SOD Assay kit-WST (FUJIFILM, Tokyo, Japan). Malondialdehyde (MDA) concentrations in liver tissue specimens were measured using a TBARS assay kit (Cayman, Chemical, Ann Arbor, MI, USA).

### 2.4. Histological Analysis

Liver tissues were fixed in 4% paraformaldehyde, embedded in paraffin, and stained with hematoxylin–eosin and Sirius Red solution. To evaluate the histopathological severity of steatohepatitis, the steatosis, activity, and fibrosis (SAF) scores were assessed for the grade of steatosis (0–3), activity (0–4), and stage of fibrosis (0–4) [34]. The percentage of atypical cells with large nuclei was quantified by counting 1300–1600 cells in the six or seven fields of view at ×100 magnification. HCCs were analyzed with the gross examination of whole livers and histologically diagnosed using sections from liver nodules. In the HFD group, liver tissues without tumors (nontumor areas) were analyzed. Histological analyses were performed under a BX43 microscope (Olympus, Tokyo, Japan). All images were captured with a DP21 digital camera (Olympus). The eyepiece lens was set to ×10 magnification and the objective lens was set to ×10 magnification to acquire images.

### 2.5. Immunoblot Analysis

Total liver homogenates were prepared as previously described [35]. Protein samples were subjected to SDS/PAGE and transferred to a PVDF membrane (Bio-Rad, Hercules, CA, USA). The corresponding primary and secondary antibodies were incubated with the membranes to visualize the protein. An antibody to hexanoyl-lysine (HEL) was obtained from JaICA (MHL-021P), and an antibody to LPS-binding protein (LBP) was obtained from Proteintech (23559-1-AP). An antibody to β-Actin was used as described previously [33]. Immunoreactive bands were densitometrically quantified and normalized to the amounts of actin present in each specimen and then averaged.

### 2.6. Real-Time Quantitative Polymerase Chain Reaction (qPCR)

Steady-state mRNA levels in the livers were determined with qPCR. Total RNA was extracted from liver specimens, followed by cDNA synthesis. qPCR was performed with Fast SYBR Green Master Mix (Thermo Fisher Scientific, Waltham, MA, USA). Primers used for this study were described previously [33]. Data were normalized to the average amounts of glyceraldehyde 3-phosphate dehydrogenase (*GAPDH*) and cyclophilin present in each sample and then averaged.

### 2.7. RNA Sequencing (RNA-Seq) Analysis

Total RNA was extracted from the liver using the NucleoSpin RNA XS kit (Macherey-Nagel, Duren, Deutschland). Sequencing was performed by Takara Bio Inc (Kusatsu, Japan) using a Smart-Seq Stranded Kit. Nucleic acid quantification was performed with electrophoresis using TapeStation or BioAnalyzer (Agilent Technologies, Santa Clara, CA, USA). Based on the position information obtained with mapping and the gene definition file, the expression levels of gene units and transcript units were calculated. The gene ontology analysis and Kyoto Encyclopedia of Genes and Genomes (KEGG) pathway analysis were performed using DAVID (The Database for Annotation, Visualization and Integrated Discovery) software. Heatmaps were constructed using ClustVis (https://biit.cs.ut.ee/clustvis/ (accessed on 10 June 2021)). The RNA-seq analysis was performed using eight mice in each group.

### 2.8. TG and FFA Concentrations and FA Composition in Liver Tissues

TG and FFA concentrations in liver tissue specimens were measured using reagent kits (FUJIFILM Wako Chemicals, Osaka, Japan). The FA composition of liver tissues was determined using a gas chromatography system. The extraction of lipids contained in the livers and the analysis of the FA composition were described previously [36].

### 2.9. Measurement of Serum and Fecal LPS Levels

LPS concentrations in serum and feces were measured with the ToxinSensor chromogenic LAL endotoxin assay kit (Genscript, Piscataway, NJ, USA), in accordance with the manufacturer’s instructions. LPS levels were expressed as endotoxin units.

### 2.10. Microbiome Analysis

The 16S rRNA gene sequencing of mice feces was performed by Takara Bio Inc (Kusatsu, Shiga, Japan). The raw sequence data from the Illumina platform were converted into forward- and reverse-read files using the FASTQ processor, and then imported into Quantitative Insights Into Microbial Ecology (QIIME), an open-source microbiome analysis platform, for further analysis. The paired-end sequences were demultiplexed, denoised, dereplicated, and merged with the Cluster Database at High Identity with Tolerance Operational Taxonomic Unit (CD-HITOTU) quality control package in QIIME. Sequence data processing, operational taxonomic unit (OTU) definition, and taxonomic assignment were performed using QIIME with a naive Bayes classifier. The differential abundance of OTUs at the different taxonomic levels, including phylum, class, family, order, genus, and species, was analyzed using the analysis of the composition of microbiomes (ANCOM). α-diversity (observed species) was determined and analyzed using Wilcoxon’s rank-sum test.

### 2.11. Statistics

Statistical analysis was conducted using IBM SPSS Statistics 26.0 (IBM, Armonk, NY, USA). Values were given as means ± standard error of the mean. When two groups were compared and an unpaired *t*-test was used for the data analysis. The chi-squared test was used to assess differences between groups for HCC incidence. A *p* value of < 0.05 was defined as statistically significant. In the HFD feeding group, the comparison was performed between the SMAPo^TN^ treatment group (tumor negative group) and nontreatment group with HCCs (tumor-positive group) to clarify the effects of SMAPo^TN^ on hepatocarcinogenesis.

The statistical analysis of α-diversity was performed with the Monte Carlo two-sample *t*-test between two unpaired groups using the α-diversity index (observed species). The diversity index used for testing was the maximum number of reads for which the diversity index could be calculated for all samples. *p*-values were corrected with the FDR method. In this analysis, because there were more than two groups, the test was performed on all two-group combinations. A *p*-value of 0.05 or less after adjustment was considered to indicate a significant difference in the diversity of the comparison groups.

## 3. Results

### 3.1. SMAPo^TN^ Is Derived and Accumulated in the Livers through the Digestive Tract without Any Side Effects

In the IVIS analysis, rhodamine-labeled SMAPo^TN^ was located in the GI tract at 1.5 h after oral administration, followed by distribution to the whole body, indicating the internalization of SMAPo^TN^ into the bloodstream even via oral administration. Rhodamine-labeled SMAPo^TN^ remained in the liver after 48 h, whereas the compound had almost cleared in other organs, indicating the SMAPo^TN^ was internalized into blood from the GI tract and mainly accumulated in the liver via the portal vein, followed by gradual spreading and excretion in other organs for 1–2 days after administration (Figure 1A).

The body weights of DKO mice fed NC or the HFD with or without SMAPo^TN^ treatment were monitored from 6 to 32 weeks of age. All mice had similar body weights at birth. After 10 weeks of age, the DKO mice fed the HFD gained weight faster and exhibited more severe obesity than DKO mice fed NC, but no differences were observed between the with or without SMAPo^TN^ groups (44.1 ± 0.9 g for DKO mice fed NC without SMAPo^TN^, 46.4 ± 0.4 g for DKO mice fed NC with SMAPo^TN^, 54.7 ± 0.4 g for DKO mice fed the HFD without SMAPo^TN^, and 52.9 ± 2.2 g for DKO mice fed the HFD with SMAPo^TN^ at 32 weeks of age; Figure 1B). The HFD feeding significantly increased liver weights and liver/body weight ratios compared with NC feeding; however, SMAPo^TN^ did not affect liver weights and liver/body weight ratios (Figure 1B).

### 3.2. SMAPo^TN^ Inhibits the Development of Liver Fibrosis and HCC

The changes in liver histology induced with the SMAPo^TN^ treatment for DKO mice fed NC or the HFD are shown in Figure 1C. To determine the histopathological severity of steatohepatitis, the SAF score was assessed (Figure 1D). In the NC feeding group, histological steatosis and the SAF score of DKO mice with SMAPo^TN^ treatment were more severe compared with those of DKO mice without the SMAPo^TN^ treatment, whereas fibrosis in DKO mice with the SMAPo^TN^ treatment was milder compared with that in DKO mice without the SMAPo^TN^ treatment. DKO mice fed the HFD showed more severe steatosis, inflammation, and fibrosis than DKO mice fed NC; however, the SMAPo^TN^ treatment suppressed steatosis and fibrosis in the livers compared with the nontreatment group (Figure 1C,D). Atypical cells with large nuclei in the nontumor area were observed in DKO mice (Figure 1E, left panel). No atypical cells were found in DKO mice fed the NC. However, the percentage of atypical cells in DKO mice fed the HFD without the SMAPo^TN^ treatment was decreased significantly due to the SMAPo^TN^ treatment (Figure 1E, right panel). As shown in Figure 1F, DKO mice often developed well-differentiated HCCs, which included atypical cells with larger nuclei (Figure 1F, blue arrows) and fat after HFD feeding for 26 weeks.

Sixteen DKO mice of forty-five fed the HFD without the SMAPo^TN^ treatment developed HCC. Conversely, no DKO mice among 10 treated with SMAPo^TN^ developed HCC (16/45, 33% vs. 0/10, 0%; *p* = 0.035, Figure 1G). Taken together with these results, the SMAPo^TN^ treatment could inhibit hepatocarcinogenesis in DKO mice. Comparing blood biochemistry analyses between DKO mice with or without the SMAPo^TN^ treatment, the HFD feeding increased AST and ALT levels compared with NC feeding in both the SMAPo^TN^ treatment and nontreatment groups, but there was no difference between these two groups (Figure 1G, left panel). The serum lipid profiles, such as TG, HDL-CHO, and LDL-CHO, were higher in DKO mice fed NC with the SMAPo^TN^ treatment than in DKO mice without the SMAPo^TN^ treatment; however, these changes were not observed in the HFD feeding groups (Figure 1H, right panel).

### 3.3. SMAPo^TN^ Treatment Reduces OS, Inflammatory Signaling, and Fibrosis-Related Factors in the Liver

Next, we investigated the effectiveness of the SMAPo^TN^ treatment in reducing OS. As shown in Figure 2A, the HFD feeding decreased GSH levels compared with NC feeding in both the SMAPo^TN^ treatment and nontreatment groups, but there was no difference between these two groups. GSSG levels did not differ between the NC and HFD groups, and the levels were higher in DKO mice fed the HFD without the SMAPo^TN^ treatment than in DKO mice treated with SMAPo^TN^. However, there was no significant change. SOD activity did not differ in DKO mice fed NC with or without the SMAPo^TN^ treatment. The HFD feeding significantly decreased SOD activity. However, the SMAPo^TN^ treatment significantly increased SOD activity (Figure 2B). The SMAPo^TN^ treatment significantly decreased the MDA levels and HEL expression in the livers of DKO mice fed NC; however, the levels did not differ in DKO mice fed the HFD (Figure 2C,D). These results suggested that SMAPo^TN^ reduced OS in the livers. Moreover, we evaluated the mRNA levels of oxidative stress-signaling genes in livers with RNA-seq analyses. A heatmap showed that the SMAPo^TN^ treatment downregulated hepatic antioxidative stress genes in both the NC and HFD feeding groups (Figure 2E). *Als2*, *Gpx7*, *Hmx1*, *Adl1*, Pdx4, Prnp, Ncf1, Pdx6, and *Txd1*, which are oxidative stress-signaling genes, were significantly suppressed with the SMAPo^TN^ treatment, mainly in the NC group (Figure 2F).

Hepatic mRNA expression levels of inflammatory cytokines (*Tnf-α* and *Il-1β*), Toll-like receptors (*Tlr*) 4, 6, and 9, and fibrosis-related genes (*Tgf-β1*, *Col1a1, and αSma*) were examined with qPCR (NC group in Figure 3A and the HFD group in Figure 3B). The mRNA levels of inflammatory cytokines and fibrosis-related genes tended to decrease, and *Tlr* 4 and 9 were significantly suppressed by the SMAPo^TN^ treatment in the NC group (Figure 3A). In the HFD group, the mRNA of *Col1a1* was significantly suppressed by the SMAPo^TN^ treatment (Figure 3B).

We also evaluated the RNA levels of hepatic inflammatory- and fibrosis-related genes in the livers using RNA-seq analyses. The heatmap showed that the SMAPo^TN^ treatment downregulated hepatic inflammatory-related signaling genes (Figure 3C) and fibrosis-related signaling genes (Figure 3D) in both the NC and HFD feeding groups. These results suggested that SMAPo^TN^ inhibited hepatic inflammation and fibrosis by reducing OS at an early stage prior to the development of NASH and hepatocarcinogenesis.

### 3.4. TG and FAs in the Livers of Mice with Steatohepatitis

The changes in lipid content and FA composition of liver tissues after the SMAPo^TN^ treatment are shown in Table 1. Consistent with liver histology (Figure 1C,D), in the NC feeding group, the TG content in the livers with the SMAPo^TN^ treatment tended to increase compared with that in the nontreatment group. In the HFD group, the TG content in the livers with the SMAPo^TN^ treatment tended to decrease compared with that in the nontreatment group, but there was no significant difference (upper part of Table 1). The FFA content in the livers did not differ between the SMAPo^TN^ treatment and nontreatment in both the NC and HFD feeding groups (upper part of Table 1). Furthermore, we measured the FFA composition in the livers (lower part of Table 1). Arachidic acid (C20:0) was slightly but significantly increased by the SMAPo^TN^ treatment in the HFD feeding group. Eicosadienoic acid (C20:2) was slightly but significantly decreased by the SMAP_O_^TN^ treatment in the HFD feeding group. However, the composition of almost all other FAs was unchanged following the SMAPo^TN^ treatment in both the NC and HFD feeding groups.

### 3.5. RNA-Seq Analyses of the Livers to Evaluate the Efficacy of SMAPo^TN^ against the Development of NASH and HCC

We examined the mechanisms underlying the changes in gene expression induced by the SMAPo^TN^ treatment by conducting RNA-seq analyses. With a mean transcripts per million (TPM) value ≥1 as the selection criterion, we selected 11,529 genes in the NC groups and 13,072 genes in the HFD groups (Table 2, upper panel). Differentially expressed genes in nontreatment mice vs. SMAPo^TN^ treatment mice were evaluated, and 10.3% of genes in the NC group and 8.2% of genes in the HFD group were altered in the livers (Table 2A). Subsequently, to further confirm the potential effects of the SMAPo^TN^ treatment on NASH and hepatocarcinogenesis, KEGG pathway analyses were performed. The top three significantly differentially expressed signaling pathways are shown in Table 2B. The metabolic (138 genes, 11.6%), biosynthesis (35 genes, 3.0%), and ER stress pathways (33 genes, 2.8%) were extracted in the NC group, and ER stress (41 genes, 3.8%), spliceosome (33 genes, 3.1%), and cancer pathways (32 genes, 3.0%) were extracted in the HFD group (Table 2B). Interestingly, the ER stress pathway was detected as a common pathway changed by the SMAPo^TN^ treatment in both the NC and HFD groups.

### 3.6. SMAPo^TN^ Treatment Downregulates ER Stress Pathway Genes

Figure 4 shows the ER stress pathway genes altered by the SMAPo^TN^ treatment in RNA-seq analyses. The SMAPo^TN^ treatment downregulated almost all ER stress pathway genes in both the NC and HFD groups (Figure 4A), and the heatmap of the ER stress pathway also showed that the SMAPo^TN^ treatment decreased RNA expression (Figure 4B). Moreover, *Atf6*, *Perk*, *Jnk*, *Edem*, and *Vip36*, which are ER stress sensors and ER stress-related genes, were significantly suppressed by the SMAPo^TN^ treatment, mainly in the HFD group (Figure 4C). In contrast, the content of TG and nonesterified FA in the livers did not differ between the NC and HFD groups with or without the SMAPo^TN^ treatment (upper part of Table 1), and the FA composition in the livers was not changed by the SMAPo^TN^ treatment (lower part of Table 1). These results suggested that SMAPo^TN^ has protective effects against NASH by reducing not only OS, but also ER stress without significant changes in FA and TG accumulation in the livers.

### 3.7. SMAPo^TN^ Treatment Downregulates Cancer Driver Genes and Cancer Pathway Genes

Hepatocarcinogenesis and HCC development from NASH are associated with changes in cancer driver genes and cancer pathway genes [10,11]. Figure 5A shows the heatmap of cancer driver genes changed by the SMAPo^TN^ treatment in RNA-seq analyses. The SMAPo^TN^ treatment downregulated HCC driver genes in the HFD group. Moreover, cancer driver genes, such as *Smb1*, *Mpk3*, *Hpb1*, *Scrp*, *Ad1b*, and *Km6a*, were significantly suppressed by the SMAPo^TN^ treatment, mainly in the HFD group, but not in the NC group (Figure 5B). In addition, the cancer pathway genes in the KEGG analyses were downregulated by the SMAPo^TN^ treatment in the HFD group (Figure 5C and Table 1), and the downregulated genes included the PI3K-Akt signaling genes (Figure 5C, lower red column). PI3K-Akt signaling genes, such as *Pik3cb*, *Kras*, *Jak1*, *Pdk1*, *Rbl2*, *Cl4a2*, *Ccnd3*, and *Fgfr3,* were significantly decreased by the SMAP_O_^TN^ treatment in the HFD group (Figure 5D, lower panel).

### 3.8. SMAPo^TN^ Treatment Decreases LPS by Improving Diversity and Changing the Microbiota Composition

LPS can affect the development of NASH and hepatocarcinogenesis in humans and DKO mice [8,9,33]. Thus, the levels of serum and fecal LPS were determined and are shown in Figure 6A. The SMAPo^TN^ treatment tended to decrease both serum and fecal LPS levels in the NC group. In the HFD group, both serum and fecal LPS levels were increased compared with those in the NC group. Moreover, the SMAPo^TN^ treatment significantly suppressed the LPS levels compared with those in the nontreatment group (Figure 6A). The protein expression levels of LBP in the livers did not differ between the NC group with or without the SMAPo^TN^ treatment, but were decreased by the SMAPo^TN^ treatment in the HFD group (Figure 6B).

To explore the effect of the SMAPo^TN^ treatment on LPS production, fecal microbiota composition was determined. The α-diversity of the fecal microbiota was decreased by the HFD feeding; however, the SMAPo^TN^ treatment significantly improved the alpha diversity in the HFD group (Figure 6C). At the phylum level, the SMAPo^TN^ treatment tended to increase the percentage of Bacteroidetes and Firmicutes in both the NC and HFD groups (Figure 6D, left panel). At the family level, the percentage of LPS-producing bacteria, such as *Ruminococcaceae*, *Porphyromonadaceae*, and *Desulfovibrionaceae,* was increased by the HFD feeding. Conversely, the HFD feeding tended to decrease *Paraprevotallaceae* in SMAP_O_^TN^ (+). Among probiotic-related bacteria, the percentage of *Lachnospiraceae* was increased, and the percentage of *Lactobacillaceae* was decreased by the HFD feeding (Figure 6D, right panel). The percentages of *Enterobacteriaceae,* which are LPS-producing bacteria, and *Streptococcaceae,* which are probiotic-related bacteria, were low (Figure 6D, right panel).

Figure 5E shows the populations of microbiota at the family level that were changed by the SMAPo^TN^ treatment compared with those in the nontreatment group. In the NC group, the populations of LPS-producing bacteria and probiotic-related bacteria did not differ between the SMAPo^TN^ treatment and nontreatment groups. In the HFD group, the SMAPo^TN^ treatment tended to decrease the populations of *Paraprevotellaceae* (−46.8%, *p* = 0.089), *Ruminococcaceae* (−6.0%, *p* = 0.781), and *Enterobacteriaceae* (−67.2%, *p* = 0.194), which are LPS-producing bacteria (Figure 6E, left panel), but these changes were not significant. In contrast, the SMAPo^TN^ treatment tended to increase the populations of *Lactobacillaceae* (+61.6%, *p* = 0.226), *Lachnospiraceae* (+9.6%, *p* = 0.586), and *Lactobacillus* (+61.5%, *p* = 0.230), which are probiotic-related bacteria (Figure 5E, right panel). These results indicated that SMAPo^TN^ might alter microbiota, such as LPS-producing and probiotic-related bacteria, reduce LPS production in feces, decrease the LPS flux into the liver, and suppress the development of NASH and hepatocarcinogenesis.

## 4. Discussion

In this study, we demonstrated that the SMAPo^TN^ treatment attenuated NASH in DKO mice fed NC (NASH model) and prevented the development of HCC in DKO mice fed a HFD (HCC model). SMAPo^TN^, which was developed using redox polymers containing antioxidant nitroxide radicals, inhibited inflammation and fibrosis in the livers through a mechanism associated with the reduction in ROS. Furthermore, we demonstrated that the SMAPo^TN^ treatment decreased ER stress-related genes, HCC driver genes, and cancer pathway genes.

A key point was that the mouse model used in this study had a metabolic profile highly relevant to human metabolic obesity, NASH, and HCC, consistent with the multiple parallel hits theory [4]. In the present work, male DKO mice spontaneously developed NASH and 10% of DKO mice developed HCCs by 50 weeks of age following NC feeding [26]. Moreover, DKO mice developed severe NASH and 33% of DKO mice developed well-differentiated HCC with severe obesity after the HFD feeding (Figure 1). DKO mice exhibited a phenotype more similar to human NASH than previous NASH models.

Several antioxidants, such as vitamin C, vitamin E, and glutathione, have not shown therapeutic effects on the development of NASH, especially in terms of clinical liver inflammation and fibrosis [17,18]. Similarly, several previously tested low-molecular-weight antioxidant compounds integrated before reaching target organs (the livers in NASH) and could not achieve the desired effects. Since SMAPo^TN^ possesses a carboxylic acid in each repeating unit, it is ionized under high pH areas, such as the small intestine, to result in internalization in the blood stream after disintegration and, finally, it accumulates in the liver for an extended period of time, as stated above.

In a previous study, another type of antioxidant nanoparticle treatment, RNP^N^, improved liver inflammation and fibrosis in a NASH model fed a choline-deficient amino acid-defined diet without obesity and HCC [30]. In the present study, we demonstrated the protective effect of the SMAPo^TN^ treatment against inflammation and fibrosis in a model with a profile similar to human NASH. Moreover, the ER stress pathway was detected as a new mechanism underlying the SMAPo^TN^ treatment effects on the development of not only NASH, but also HCC. Recent studies have shown that the three branches of the unfolded protein response in ER stress, including PERK, IRE1α, and ATF6, cross-talk with the transcription factor NF-κB, activate inflammatory signaling pathways (e.g., TNFα), and further recruit JNK to activate cell apoptosis [37], with ER stress influencing hepatocarcinogenesis [7]. In this study, the SMAPo^TN^ treatment decreased *Perk* in the NC group and *Atf6*, *Perk*, *Jnk*, *Edem,* and *Vip36* in the HFD feeding group. These results indicated that the SMAPo^TN^ treatment suppressed ER stress and OS and prevented the development of NASH and HCC.

HCC is induced by hepatic inflammation and fibrosis in NASH [3], and several cancer driver genes and cancer pathway genes are associated with hepatocarcinogenesis [10,11]. In the present study, a critical new finding was that SMAPo^TN^ decreased multiple HCC driver genes and cancer pathway genes (Figure 5). Due to these decreases being mainly observed in the HFD group, SMAPo^TN^ could indirectly suppress HCC by reducing inflammation and fibrosis in the liver. Nevertheless, our observation that SMAPo^TN^ as antioxidants could suppress the development of HCC was not only of scientific interest, but also of potential clinical relevance, because there is currently no treatment for the prevention of HCCs.

In addition, PI3K-Akt-mTOR signaling genes in cancer pathways were significantly decreased by the SMAPo^TN^ treatment in the HFD group. β-catenin, p53-Rb, and PI3K-Akt-mTOR pathways are major signaling pathways involved in genetic alterations in HCC [10,38]. PI3K activates the Akt signaling cascade, and Akt signaling plays a critical role in regulating diverse cellular functions, including metabolism, growth, proliferation, transcription, and protein synthesis, in both cancer cells and healthy cells [39]. The significant influence of SMAPo^TN^ on PI3K-Akt-mTOR signaling observed in the HFD group indicated that the SMAPo^TN^ treatment might not directly affect PI3K signaling, but suppress the development of HCC induced by NASH by improving metabolic pathways, including insulin signaling associated with PI3K-Akt.

In the present study, we demonstrated that the SMAPo^TN^ treatment decreased LPS levels and LPS flux into the liver of mice in the HFD group (Figure 6A,B). LPS from microbiota accelerates the development of NASH and HCC [4,6,8], and the inhibition of LPS suppresses the development of NASH [40]. Because some LPS-producing bacteria tended to decrease after the SMAPo^TN^ treatment, it is possible that blood LPS and fecal LPS levels subsequently decreased, leading to a protective effect on the liver. Dysbiosis contributes to liver fibrogenesis by decreasing the diversity of microbiota and increasing the LPS flux from the intestines [41]. The SMAPo^TN^ treatment improved the α-diversity and changed the composition of microbiota, including LPS-producing and probiotic-related bacteria (Figure 6C,D). In a previous study, a pH-insensitive RNP^O^ treatment inhibited the increase in commensal bacteria by scavenging ROS [29]. We also found that the SMAPo^TN^ treatment might increase some probiotic-related bacteria, such as lactate-producing *Lactobacillaceae*, which are known to protect intestinal barrier function [42]. Previous studies have reported that probiotic-related bacteria, including *Lactobacillus* and *Bifidobacterium,* increased the diversity of gut microbiota and decreased LPS, thereby reducing liver inflammation [43,44]. The results of the current study indicated that the SMAPo^TN^ treatment had protective effects against not only NASH, but also HCC through the gut–liver axis by improving the gut microbiota diversity and decreasing the LPS flux from the intestines.

## 5. Conclusions

The oral administration of SMAPo^TN^ attenuated NASH development and inhibited hepatocarcinogenesis in DKO mice. These benefits were attributed to the dual effects of SMAPo^TN^ on the improvements in ER stress in the liver and microbiota in the intestine. SMAPo^TN^ may be a new therapeutic option for not only NASH subjects, but also those with a high risk of HCC.

## Figures and Tables

**Figure 1 antioxidants-11-01939-f001:**
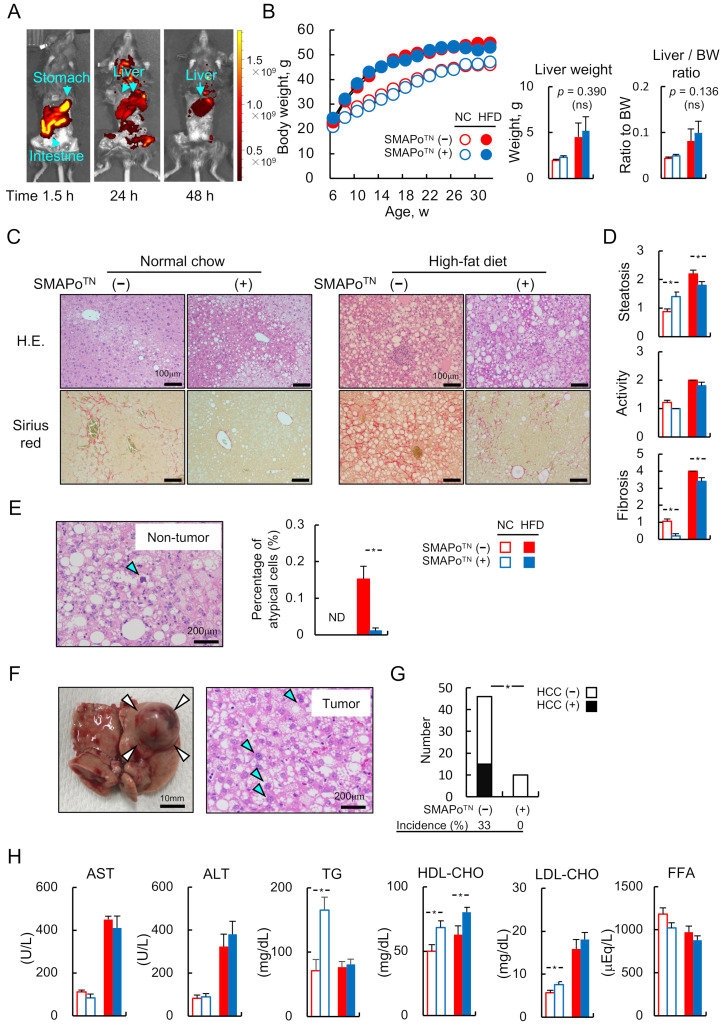
SMAPo^TN^ treatment suppresses liver fibrosis and hepatocellular carcinoma without body weight changes and side effects. (**A**) Rhodamine-labeled SMAPo^TN^ was detected using intelligent visual information system (IVIS) analysis from 1.5 h to 48 h after administration. The color scale indicates the radiant efficiency of rhodamine-labeled SMAPo^TN^ with the IVIS (p/s/cm^2^/sr/µ/cm^2^). (**B**) Time course of changes in body weight during the experimental period from 6 weeks of age to 32 weeks of age (left panel) and liver weight (center panel) and liver/body ratio (right panel) at the end of the experiment in DKO mice fed NC or HFD with or without SMAPo^TN^ treatment (n = 10–13/group). (**C**) Histopathology of steatohepatitis in DKO mice fed NC or HFD with or without SMAPo^TN^ treatment. Hematoxylin and eosin (HE)-stained sections (upper panels) and Sirius-Red-stained sections (lower panels) of liver specimens at 32 weeks of age (scale bar, 100 µm). (**D**) The steatosis (left), activity (center), and fibrosis (right) (SAF) scores (n = 10–13/group). (**E**) HE-stained sections of atypical cells with large nuclei in the nontumor area in DKO mice fed the HFD without SMAPo^TN^ treatment (left panel, blue arrow) and the percentage of atypical cells (right panel, n = 10–13/group). (**F**) A macroscopic view of hepatocellular carcinoma (HCC) in DKO mice fed HFD without SMAPo^TN^ treatment (left panel, white arrows) and HE-stained section (right panel, blue arrows show cancer cells with larger nuclei). (**G**) Comparison of tumor numbers between SMAPo^TN^ treatment and non-SMAPo^TN^ groups. Sixteen DKO mice of forty-five fed the HFD without SMAPo^TN^ treatment developed HCC. Conversely, no DKO mice among 10 treated with SMAPo^TN^ developed HCC. (**H**) Analysis of blood biochemistry (aspartate aminotransferase (AST), alanine aminotransferase (ALT), triglyceride (TG), high-density lipoprotein cholesterol (HDL-CHO), low-density lipoprotein cholesterol (LDL-CHO), and free fatty acid (FFA)) in DKO mice fed NC or HFD with or without SMAPo^TN^ treatment at 32 weeks of age (n = 10–13/group). Error bars represent SEM. ND; not determined; * *p* < 0.05 indicates a significant difference between DKO mice with or without SMAPo^TN^ treatment.

**Figure 2 antioxidants-11-01939-f002:**
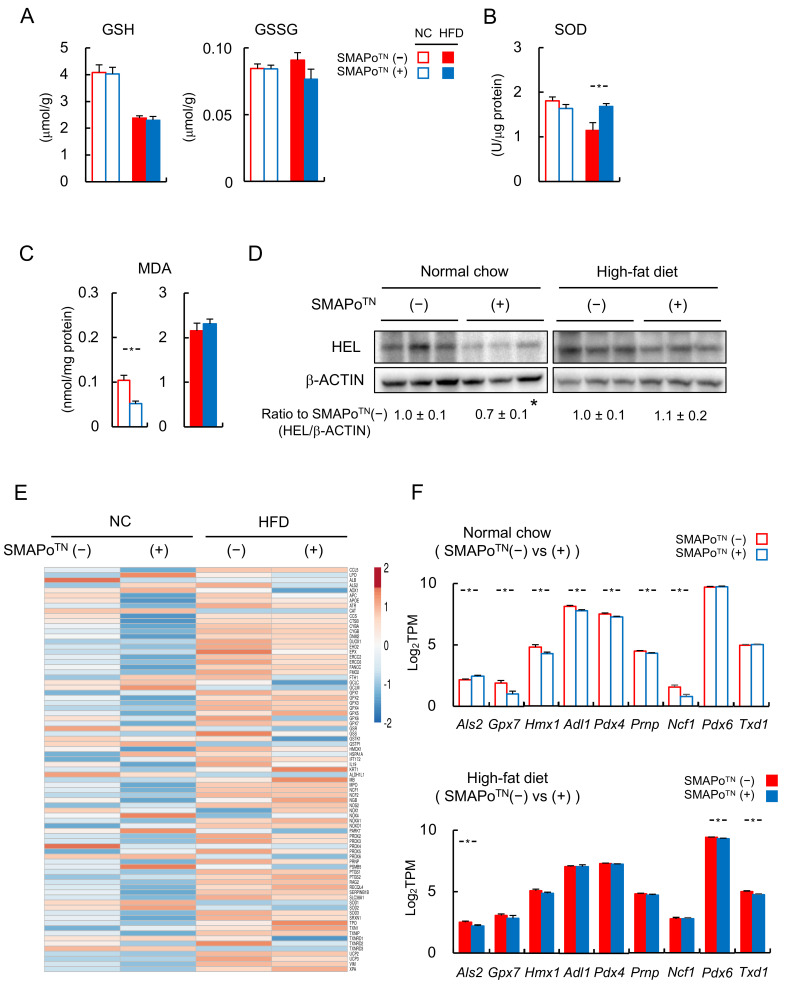
SMAPo^TN^ treatment suppresses oxidative stress. (**A**) GSSG/GSH concentrations in the livers of DKO mice fed the NC or HFD with or without SMAPo^TN^ treatment at 32 weeks of age (n = 10/group). (**B**) SOD activity in the livers of DKO mice fed the NC or HFD with or without SMAPo^TN^ treatment at 32 weeks of age (n = 10/group). (**C**) Malondialdehyde (MDA) concentrations in the livers of DKO mice fed the NC or HFD with or without SMAPo^TN^ treatment at 32 weeks of age (n = 10/group). (**D**) Immunoblot analysis of hexanoyl-lysine (HEL) in the livers of mice fed the NC or HFD with or without SMAPo^TN^ treatment at 32 weeks of age. β-Actin was used as a loading control. The data below the band show the ratio of HEL/β-actin normalized to that of SMAPo^TN^ in NC or HFD groups. (**E**) Heatmap of 80 genes involved in oxidative stress signaling in the livers of DKO mice fed the NC or HFD with or without SMAPo^TN^ treatment analyzed with RNA-seq. (**F**) Histograms showing the expression of oxidative stress-related genes that were significantly downregulated with SMAPo^TN^ treatment in NC or HFD groups (n = 8/group). Error bars represent SEM. * *p* < 0.05 indicates a significant difference between DKO mice with or without SMAPo^TN^ treatment.

**Figure 3 antioxidants-11-01939-f003:**
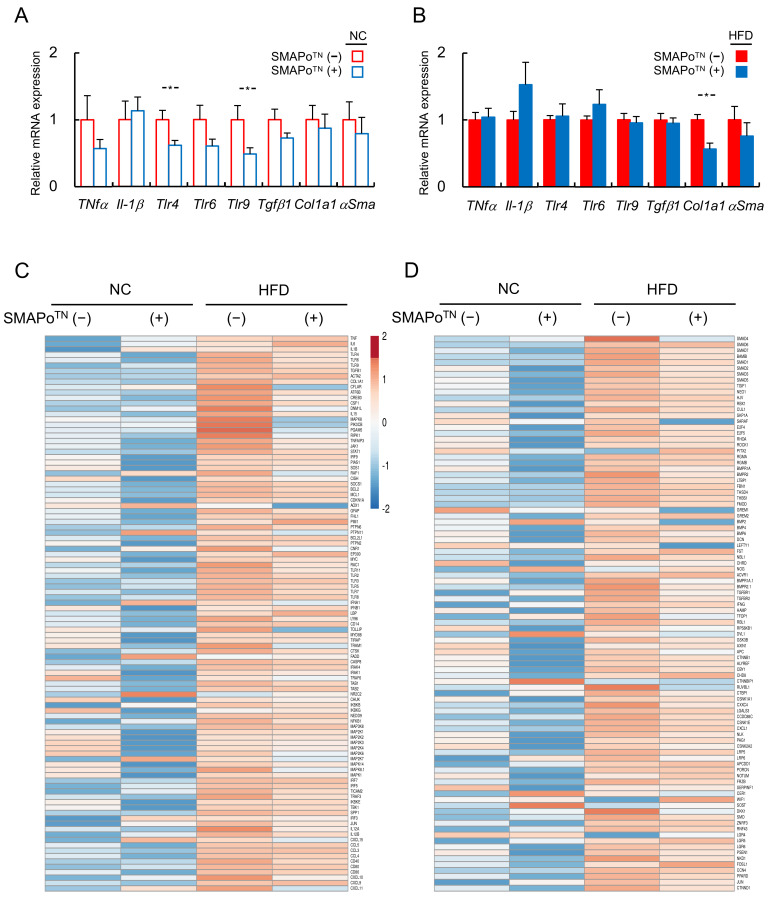
SMAPo^TN^ treatment suppresses activation of hepatic inflammatory and fibrotic signaling. qPCR analyses of inflammatory cytokines (*Tnfα* and *Il1β*), Toll-like receptor (*Tlr*) *4*, *6*, and *9*, and fibrosis-related genes (*Tgfβ1,*
*Col1a1,* and *αSma*) in the livers of mice fed NC (**A**) or HFD (**B**) (n = 10/group). A heatmap of 104 genes involved in inflammatory signaling (**C**) and a heatmap of 94 genes involved in fibrosis signaling (**D**) in the livers of DKO mice fed NC or HFD with or without SMAPo^TN^ treatment analyzed with RNA-seq are shown (n = 8/group). Error bars represent SEM. * *p* < 0.05 indicates a significant difference between DKO mice with or without SMAPo^TN^ treatment.

**Figure 4 antioxidants-11-01939-f004:**
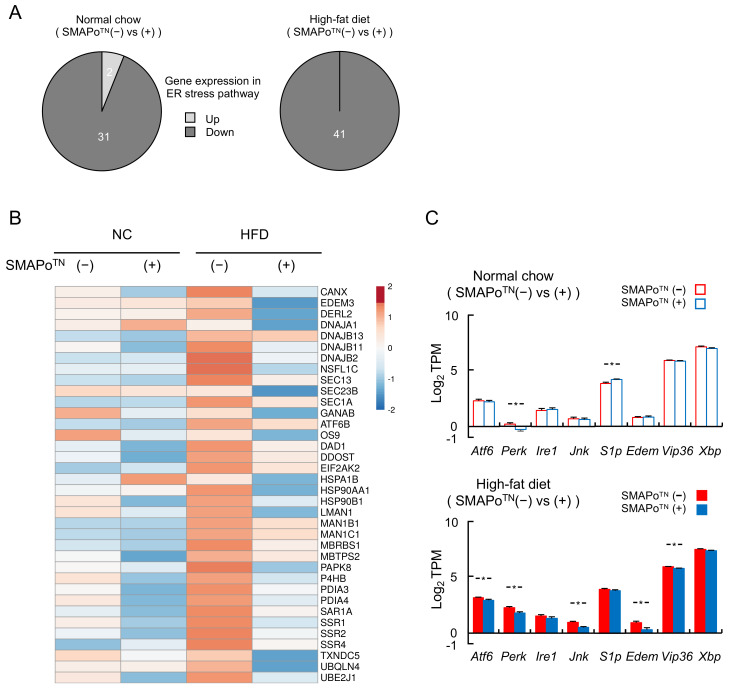
SMAPo^TN^ administration reduces the expression of ER stress-response-related genes. (**A**) The number of genes upregulated or downregulated by SMAPo^TN^ treatment in the ER stress pathway analyzed with RNA-seq. (**B**) A heatmap of 36 genes involved in the ER stress pathway analyzed with RNA-seq. (**C**) Histograms of gene expression levels in the representative ER stress pathway of the liver analyzed with RNA-seq (n = 8/group). Gene expression levels were shown by the mean transcripts per million (TPM). Error bars represent SEM. * *p* < 0.05 indicates a significant difference between DKO mice with or without SMAPo^TN^ treatment.

**Figure 5 antioxidants-11-01939-f005:**
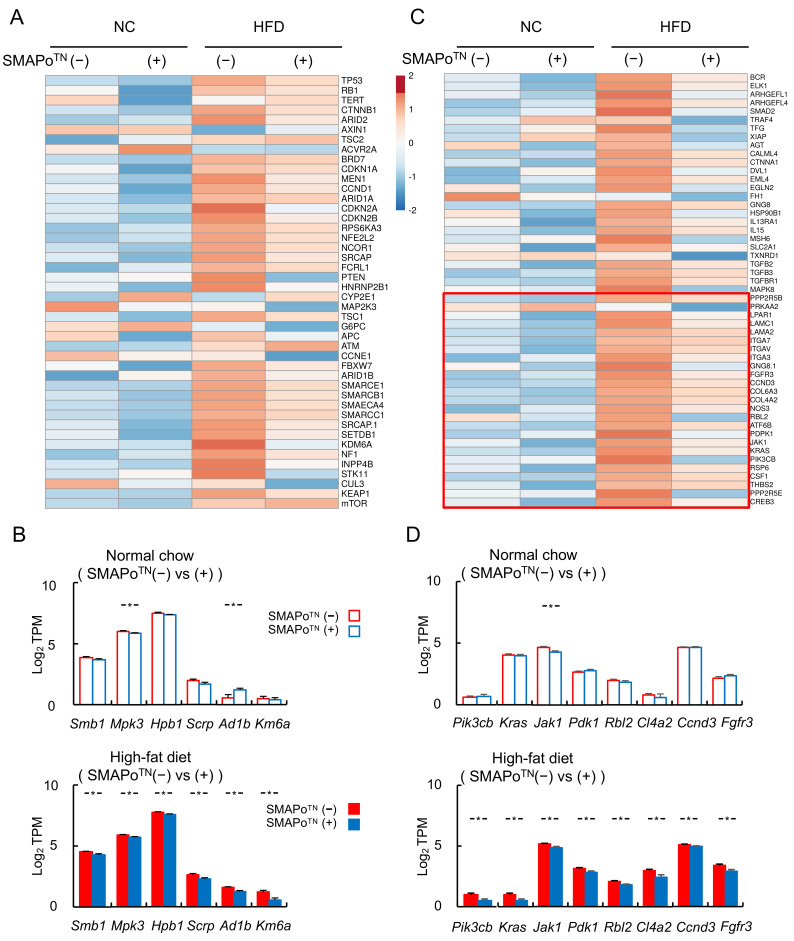
SMAPo^TN^ treatment reduces the expression of cancer driver genes and cancer pathway-related genes. (**A**) A heatmap of 44 cancer driver genes analyzed with RNA-seq. (**B**) Histograms showing the expression of cancer driver genes that were significantly downregulated by SMAPo^TN^ treatment in HFD groups (n = 8/group). (**C**) A heatmap of 51 cancer pathway-related genes in KEGG. The lower red column shows PI3K-Akt signaling pathway-related genes. (**D**) Histograms showing the expression of cancer pathway-related genes that were significantly downregulated by SMAPo^TN^ treatment in HFD groups (n = 8/group). Error bars represent SEM. * *p* < 0.05 indicates a significant difference between DKO mice with or without SMAPo^TN^ treatment.

**Figure 6 antioxidants-11-01939-f006:**
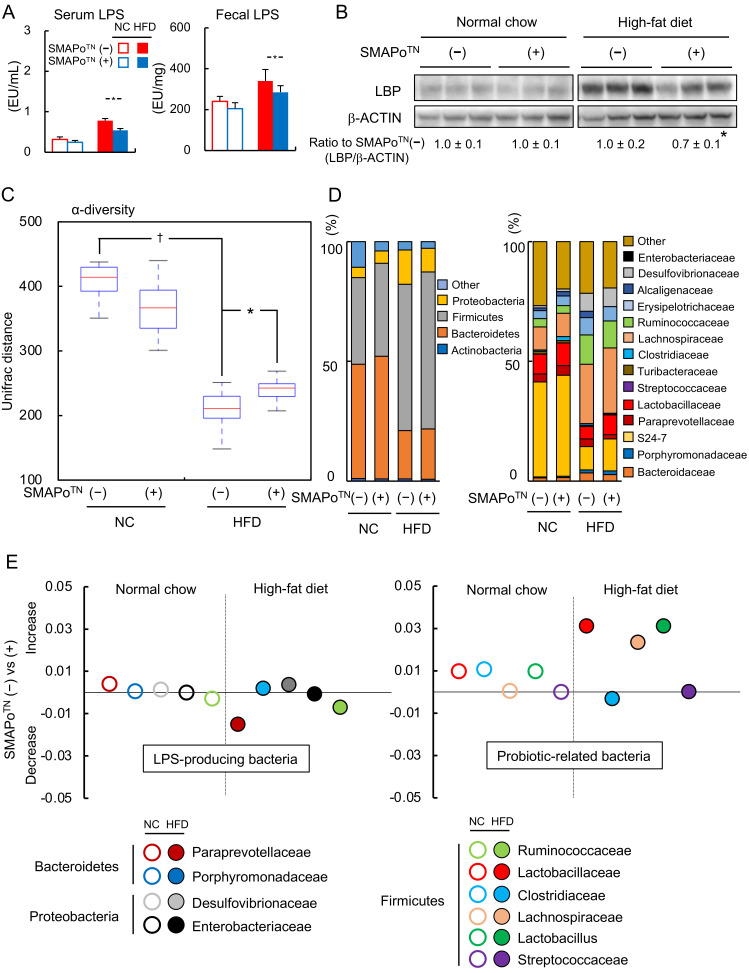
Effects of SMAPo^TN^ treatment on lipopolysaccharide (LPS) and gut microbiota in DKO mice. (**A**) Serum (left panel) and fecal (right) LPS concentrations in DKO mice fed NC or HFD with or without SMAPo^TN^ treatment at 32 weeks of age (n = 10/group). (**B**) Immunoblot analysis of LPS-binding protein (LBP) in the livers of DKO mice fed NC or HFD with or without SMAPo^TN^ treatment at 32 weeks of age. The β-actin bands were used as loading controls. The data below the band show the ratio of LBP/β-actin normalized by SMAPo^TN^ in NC or HFD groups. (**C**) Changes in α-diversity in gut microbiota (n = 10/group). (**D**) Relative abundances of predominant bacteria at the phylum level (left panel) and the family level (right panel). (**E**) Comparative analysis of the taxonomic composition of the microbial community at the family and higher levels. The left panel shows LPS-producing bacteria and the right panel shows probiotic-related bacteria. Error bars represent SEM. * *p* < 0.05 indicates a significant difference between DKO mice with or without SMAPo^TN^ treatment. ^†^
*p* < 0.05 indicates a significant difference between DKO mice fed NC or HFD.

**Table 1 antioxidants-11-01939-t001:** Lipid content and FFA composition in liver tissues.

	Normal Chow						High-Fat Diet					
	SMAPo^TN^ (−)			SMAPo^TN^ (+)			SMAPo^TN^ (−)			SMAPo^TN^ (+)		
	(n = 8)			(n = 8)			(n = 8)			(n = 8)		
TG content	94.93	±	18.31	117.13	±	8.71	137.57	±	7.39	112.75	±	10.88
(μmol/g liver)												
FFA content	8.51	±	1.16	10.60	±	1.14	10.44	±	0.50	9.65	±	0.91
(μmol/g liver)												
FFA composition (mg/g liver)												
C6:0		ND			ND			ND			ND	
C8:0		ND			ND			ND			ND	
C10:0		ND			ND			ND			ND	
C11:0		ND			ND			ND			ND	
C12:0		ND			ND		0.03	±	0.01	0.04	±	0.01
C13:0		ND			ND			ND			ND	
C14:0	0.64	±	0.14	0.73	±	0.11	0.64	±	0.07	0.74	±	0.10
C14:1	0.04	±	0.00	0.05	±	0.01	0.04	±	0.01	0.06	±	0.01
C15:0	0.1	±	0.0	0.1	±	0.0	0.1	±	0.0	0.1	±	0.02
C15:1		ND			ND			ND			ND	
C16:0	29.12	±	5.84	30.23	±	3.95	28.82	±	2.95	25.65	±	3.90
C16:1	6.66	±	1.75	7.75	±	1.45	5.16	±	0.67	5.11	±	0.86
C17:0	1.43	±	1.34	0.10	±	0.01	0.11	±	0.02	0.14	±	0.02
C17:1		ND			ND			ND			ND	
C18:0		ND			ND			ND			ND	
C18:1n9	49.04	±	12.75	45.02	±	13.05	57.16	±	10.46	29.43	±	10.22
C18:2n6	12.68	±	3.43	17.69	±	2.44	6.54	±	0.55	7.92	±	0.87
C18:3n6	0.21	±	0.05	0.21	±	0.03	0.16	±	0.02	0.13	±	0.02
C18:3n3	0.53	±	0.10	0.53	±	0.06	0.25	±	0.02	0.27	±	0.04
C20:0	0.26	±	0.07	0.20	±	0.03	0.06	±	0.00	0.15	±	0.05 *
C20:1n9	1.41	±	0.26	1.34	±	0.21	2.00	±	0.25	1.80	±	0.28
C20:2	0.23	±	0.03	0.24	±	0.02	1.05	±	0.10	0.77	±	0.08 *
C20:3n6	0.46	±	0.09	0.45	±	0.06	0.41	±	0.04	0.40	±	0.03
C20:4n6		ND			ND			ND			ND	
C20:3n3		ND			ND			ND			ND	
C20:5n3	0.26	±	0.06	0.26	±	0.04	0.09	±	0.01	0.09	±	0.01
C21:0	1.45	±	0.12	1.40	±	0.11	1.80	±	0.17	2.04	±	0.08
C22:0		ND			ND			ND			ND	
C22:1n9		ND			ND			ND			ND	
C22:2		ND			ND		0.35	±	0.04	0.24	±	0.03
C22:6n3	2.47	±	0.36	2.36	±	0.17	0.88	±	0.13	0.98	±	0.12
C23:0		ND			ND			ND			ND	
C24:0		ND			ND			ND			ND	
C24:1n9		ND			ND			ND			ND	

Values are presented as the group means ± SEM (n = 8). ND; not determined; * *p* < 0.05, between DKO mice with or without SMAP_O_^TN^ treatment.

**Table 2 antioxidants-11-01939-t002:** Gene expression changes induced by SMAPo^TN^ treatment in RNA-seq analysis. (**A**)Number of genes that had a mean TPM above 1 and were significantly upregulated or downregulated (*p* < 0.05) by SMAPo^TN^. (**B**) Treatment in NC or HFD groups (n = 8/group). Top 3 genes in KEGG pathway analysis.

A								
Group	Normal Chow				High-Fat Diet			
Gene number (TPM ≥ 1)	11,529				13,072			
Differentially expressed	Up	Down	Total	%	Up	Down	Total	%
SMAP_O_^TN^ (−) vs. (+)	167	1019	1186	10.3	40	1037	1077	8.2
**B**								
**Group**	**Normal Chow** **(SMAP_O_^TN^ (−) vs. (+))**						**High-Fat Diet** **(SMAP_O_^TN^ (−) vs. (+))**	
Variable number of pathway	42			Variable number of pathway			28	
Pathway (KEGG, top three)	Genes (n)		%	Pathway (KEGG, top three)			Genes (n)	%
Metabolic pathways	138		11.6	Endoplasmic reticulum			41	3.8
Biosynthesis	35		3.0	Splicesome			33	3.1
Endoplasmic reticulum	33		2.8	Cancer pathway			32	3.0

## Data Availability

Not applicable.

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
