# Peer review of "Antioxidative Self-Assembling Nanoparticles Attenuate the Development of Steatohepatitis and Inhibit Hepatocarcinogenesis in Mice"

_antioxidants, 2022, doi:10.3390/antiox11101939_

Round 1
Reviewer 1 Report
Thanks to the authors for the work done in identifying a possible treatment for the liver in NASH and hepatocellular carcinoma.
Some comments:
1. Line 89…. However, most of the conventional antioxidants did not achieve suitable effects. These conventional antioxidants spread throughout the whole body and are eliminated rapidly. In addition, these antioxidants cause dysfunctions of important redox reactions in normal cells including an electron transport chain. What are the conventional antioxidants? What are the dysfunctions of important redox reactions? Add references or explanations, please.
2. Fig1A- the scale is not visible.
Fig. 1A 24 hours after administration of Rhodamine-labeled SMAPoTN. Are labeled SMAPoTNs detected throughout the mouse body (lungs, muscles, heart)? After 48 hours labeled SMAPoTNs detected mainly in the liver. Can you give any suggestions as to what happens to SMAPoTNs during this time?
Fig1D-there are no markings for the scatter of the mean value error on the graph.
3. Line 453. SMAPoTN, which was developed using redox polymers containing antioxidant nitroxide radicals, inhibited inflammation and fibrosis in the liver through a mechanism associated with a reduction in ROS. What exactly is the mechanism? The authors gave studies of lipid peroxidation, without determining ROS/antioxidant enzyme activity. Oxidative stress involves an imbalance between ROS production and ROS utilization. Only the assessment of lipid peroxidation is presented here. What exactly takes place in these diseases: inhibition of antioxidant enzyme/antioxidant activity or excess ROS production?
Reviewer 2 Report
The accumulation of fat in the liver is one of the causes of chronic liver disease. Non-alcoholic fatty liver disease (NAFLD) is becoming more common worldwide, affecting approximately 25% of the population. NAFLD can progress to non-alcoholic steatohepatitis (NASH), an aggressive form of fatty liver disease, which is marked by liver inflammation and may progress to cirrhosis and liver failure. In this work, Watahiki et al. report that the treatment with antioxidative self-assembling nanoparticles (SMAPoTN) prevent –observing a reduction of both ER stress in liver cells and lipopolysaccharide production by intestinal microbiota– the development of NASH and its associated hepatocarcinogenesis in a p62 and Nrf2 double knockout mice model. Overall, the data is acceptable, however, there are several issues that must be addressed.
Major issues:
Figure 1E. Polyploid cells are normal in the aged liver. For comparisons, include a representative H.E. image at same magnification of HFD SMAPoTN (+) group and quantify the percentage of atypical cells with larger nuclei in the 4 groups (DKO mice fed NC and HFD with or without SMAPoTN, respectively)
Figure 1E. Authors compare 10-13 animals/group, in results they mention a 16/45, 33% vs. 0/10 comparison. Clarify
Figure 1G. There are no significant differences in TG levels between NC -/+ SMAPoTN groups. Double-check the data/statistical test (TG, and HDL-CHO levels between -/+ SMAPoTN in NC and HFD groups, respectively). The same applies to Figure 2D; Tlr6 seems to be increased in + SMAPoTN (+)
Figure 2. OS reduction by SMAPoTN is not convincing based on current data (see comment below). Additional evidence (e.g., GSH/GSSG levels and SOD activity) must be shown
Figure 3 and 4. The downregulation of some ER stress response-related, cancer driver, and cancer pathway-related genes is not clear. Mean values of Vip36 (HFD) in 3C, Mpk3 (NC or HFD) and Hpb1 (HFD) in 4B, and Ccnd3 (HFD) in 4D are almost the same in (-) and (+). The reported significant differences between similar mean values with SEM near 0 should be excluded. Include the number of samples analyzed
Table 1. Data shown is inconsistent with Figure 1C and D. Not significant differences were observed in TG content/liver after SMAPoTN treatment between the NC or HFD groups. Double-check the significance for C20:0. C20:2 is not increased in SMAPoTN (+) HFD group
Minor issues:
Material and Methods
Briefly describe the IVIS system and method/analysis
Include the method used for the measurements of AST, ALT, TG, HDL- and LDL-CHO and FFA
Include the imaging setup for image acquisition (Figures 1C, 1E) including microscope, lens, and software
Figure 1E. Include scale bar (left), bar sizes, and labels (right image). Lines 52, 109; define p62/Sqstm1
Figure 2A. Expand the Y axis scale (0-0.3) for better visualization of MDA levels and SEM.
Figure 5. Include the alpha-diversity analysis in methods section 2.11. Double-check numbers in results (lines 428, 429, and 432)
Table 2. Split A and B more clearly. Double-check numbers (Total) in SMAPoTN (−) vs. (+) HDF comparison
Line 77; Europe is not a country
Line 137; standard chow => normal chow
Lines 143; 145; include MW units
Line 143; a-methyl-o-hydroxy => alpha-Methoxy-omega-hydroxy
Lines 247, 248; when mentioned in text, include significance (p values) and not significant (ns) in graphs.
Line 294; indicate => suggest
Line 341; transcripts per million value => transcripts per million (TPM) value
Line 419; Paraprevotallaceae seem to be reduced by HFD feeding in SMAPoTN (+)
Typo & text errors
Line 143; co polymer => copolymer
Line 241; 48 => 32
Lines 300, 302; (Figure 2D left panel) => (Figure 2C), 2E => 2D
Table 1 upper part, line 328; FA => FFA
Round 2
Reviewer 1 Report
Many thanks to the authors for the detailed elaboration of the comments to the manuscript!!!
Reviewer 2 Report
The authors have reasonably addressed the issues raised in my earlier review.